# Influence of Halides on Elastic and Vibrational Properties of Mixed-Halide Perovskite Systems Studied by Brillouin and Raman Scattering

**DOI:** 10.3390/ma16113986

**Published:** 2023-05-26

**Authors:** Furqanul Hassan Naqvi, Syed Bilal Junaid, Jae-Hyeon Ko

**Affiliations:** School of Nano Convergence Technology, Nano Convergence Technology Center, Hallym University, Chuncheon 24252, Republic of Korea; furqanhassan05@gmail.com (F.H.N.); syedbilaljunaid@yahoo.com (S.B.J.)

**Keywords:** mixed lead halide perovskites, MAPbBr_3−*x*_Cl*_x_*, Raman spectroscopy, Brillouin spectroscopy

## Abstract

The relationship between halogen content and the elastic/vibrational properties of MAPbBr_3−*x*_Cl*_x_* mixed crystals (*x* = 1.5, 2, 2.5, and 3) with MA = CH_3_NH_3_^+^ has been studied using Brillouin and Raman spectroscopy at room temperature. The longitudinal and transverse sound velocities, the absorption coefficients and the two elastic constants *C*_11_ and *C*_44_ could be obtained and compared for the four mixed-halide perovskites. In particular, the elastic constants of the mixed crystals have been determined for the first time. A quasi-linear increase in the sound velocity and the elastic constant *C*_11_ with increasing chlorine content was observed for the longitudinal acoustic waves. *C*_44_ was insensitive to the Cl content and very low, indicating a low elasticity to shear stress in mixed perovskites regardless of the Cl content. The acoustic absorption of the LA mode increased with increasing heterogeneity in the mixed system, especially for the intermediate composition where the Br and Cl ratio was 1:1. In addition, a significant decrease in the Raman-mode frequency of the low-frequency lattice modes and the rotational and torsional modes of the MA cations was observed with decreasing Cl content. It clearly showed that the changes in the elastic properties as the halide composition changes were correlated with the lattice vibrations. The present findings may facilitate a deeper understanding of the complex interplay between halogen substitution, vibrational spectra and elastic properties, and may also pave the way for optimizing the operation of perovskite-based photovoltaic and optoelectronic devices by tailoring their chemical composition.

## 1. Introduction

Mixed-lead-halide perovskites have emerged as a promising group of materials for various optoelectronic applications, particularly in the field of photovoltaics [1]. These materials exhibit unique properties such as bandgap tunability, high carrier mobilities and long carrier lifetimes, making them highly attractive for solar cell applications [2,3,4]. Perovskite-based solar cells have rapidly improved their efficiency from just 3.8% in 2009 to over 25% in recent years, rivalling conventional silicon-based solar cells [5,6]. Their ability to continuously tune the band gap, combined with high carrier lifetimes and mobilities, solution processability and improved stability, make mixed-halide perovskites an attractive candidate for next-generation solar cell technologies [7,8]. In addition, mixed perovskite materials offer versatility through their compositional flexibility, allowing researchers to tailor their properties according to the desired application. Despite the numerous advantageous properties of perovskites, their long-term stability remains a significant limitation that requires further investigation [9].

Brillouin and Raman spectroscopy are crucial techniques for the investigation of perovskite materials, as they provide a comprehensive analysis of structural, vibrational and mechanical properties [10,11]. Raman spectroscopy provides critical insight into the crystal structure and phase transitions, as well as the vibrational modes that affect the performance of perovskite devices. Meanwhile, room-temperature Brillouin spectroscopy is essential for determining the elastic properties that could optimize perovskite materials for device stability and performance [12]. Furthermore, elastic constants are fundamental physical properties that are useful barometers for checking the reliability of theoretical calculations. By combining these techniques, we can comprehensively analyze mixed perovskite materials and optimize their properties, leading to better device performance.

In this study, we present a room-temperature Brillouin and Raman spectroscopic investigation of mixed-lead-halide perovskites, MAPbBr_3−*x*_Cl*_x_* with *x* = 1.5 to 3, to gain insight into their vibrational properties and structural stability. We study the variations in optical-mode and acoustic-mode frequencies as a function of halide composition, where the chlorine content increases from 1.5 to 3 at the X-site anion in the ABX_3_ perovskite structure. Previous studies have investigated the effect of chlorine substitution at the X site on MAPbI_3−*x*_Cl*_x_* systems. A recent study demonstrated an increase in the photovoltaic performance and stability of MAPbI_3−*x*_Cl*_x_* mixed perovskites with increasing Cl content [13]. Another study observed increased photoexcitation lifetimes at higher Cl contents [14]. Density functional theory (DFT) modelling by Silvia et al. showed that the incorporation of Cl into the solar-cell device improved the charge collection efficiency [15,16]. Comparing two systems, MAPbI_3−*x*_Cl*_x_* and MAPbI_3−*x*_Br*_x_*, Tombe et al., discovered that Cl-based mixed perovskites exhibited superior film morphology and photovoltaic efficiency [17]. In addition, Zheng et al. highlighted that the charge diffusion properties of MAPbBr_3−*x*_Cl*_x_* are superior to those of the MAPbI_3−*x*_Cl*_x_* system [18]. While there are numerous reports on MAPbI_3−*x*_Cl*_x_* systems, the MAPbBr_3−*x*_Cl*_x_* system remains underexplored. A limited number of studies have investigated the stability, vibrational and photovoltaic properties for this system [19,20,21,22,23]. Furthermore, information on the elastic properties of these materials remains scarce in the existing literature.

In the present work, we have used Brillouin spectroscopy to assess the elastic properties of these materials and to investigate the influence of Cl addition at the X site. The acoustic phonon velocities and the corresponding elastic constants were found to increase with Cl content, suggesting a more rigid and stable lattice structure. We further correlated the behavior of the acoustic phonon modes with the optical phonon modes probed by Raman spectroscopy. The measurement range was extended to a lower frequency range compared to our previous Raman study [21]. Low-frequency Raman modes corresponding to lattice vibrations showed an increase in wavenumber with higher Cl content. This observation indicated a change in the bond strength within the lattice due to the replacement of weaker Pb-Br bonds by stronger Pb-Cl bonds upon addition of Cl.

## 2. Experimental Methods

### 2.1. Chemicals

Lead chloride (PbCl_2_, 99.999%), diethyl ether (HPLC grade, 99.9%), lead bromide (PbBr_2_, >98%), hydrochloric acid (HCl, 37%, ACS reagent), hydrobromic acid (48%, ACS reagent), ethanol (anhydrous 99.5%), dimethyl sulfoxide (DMSO, anhydrous 99.9%), *N*,*N*-dimethylformamide (DMF, anhydrous 99.8%) and methylamine (CH_3_NH_2_, 40% in water) were purchased from Sigma Aldrich (Saint Louis, MO, USA). Methylammonium chloride (MACl) and methylammonium bromide (MABr) were synthesized using above precursors, details of which are mentioned in our previous study [10]. 

### 2.2. MAPbBr_3−x_Cl_x_ Single-Crystal Synthesis

The single crystals of MAPbBr_3−*x*_Cl*_x_* with varying degrees of chlorine substitution (*x* = 1.5, 2, 2.5 and 3) were grown using the solvent evaporation method. This method involved mixing equimolar ratios of MACl and MABr with PbCl_2_ and PbBr_2_, respectively, in their respective solvents. The solvent used was a blend of *N*,*N*-dimethylformamide (DMF) and dimethyl sulfoxide (DMSO) with different volume ratios for various compositions. After complete dissolution of the powders in solvent, the solution was filtered by using a 0.2 μm syringe filter, and the filtrate was kept at constant temperature (80–100 °C) for one or two days until the crystals started to appear. The crystals were then washed with dichloromethane and dried at 60 °C for one night in an oven.

### 2.3. X-ray Diffraction (XRD)

Powder X-ray diffraction (XRD) patterns were obtained at room temperature using a high-resolution powder X-ray diffractometer (PANalytical X’pert Pro MPD, Malvern, UK). Cu Kβ radiation (λ = 1.5406 Å) was employed to probe the samples within a scan range of 10° < 2Θ < 60° at a scan rate of 3° per minute. Crystalline powders were prepared for the measurements by crushing single crystals.

### 2.4. Brillouin Spectroscopy

A tandem, multi-pass Fabry-Perot interferometer (TFP-2, JRS Co., Zürich, Switzerland) was utilized to record room-temperature Brillouin spectra, with an excitation source of wavelength 532 nm. Measurements were conducted using backscattering geometry, facilitated by a modified microscope (BH-2, Olympus, Tokyo, Japan) in the backscattering geometry $c\left(a,a+b\right)\bar{c}$, where *a*, *b* and *c* represent the cubic axes. In this geometry, the scattered light was concurrently collected along the same path as that of the incident light.

### 2.5. Raman Spectroscopy

Raman measurements were conducted using a standard Raman spectrometer (LabRam HR800, Horiba Co., Kyoto, Japan). Room-temperature spectra were obtained in a frequency range spanning from 10 to 3500 cm^−1^. The Raman spectrometer was equipped with a low-frequency notch filter, enabling the lowest frequency limit of 10 cm^−1^. A diode-pumped solid-state laser with a wavelength of 532 nm was employed to excite the single crystals. All measurements were performed utilizing an optical microscope (BX41, Olympus, Tokyo, Japan) with a 50× magnification objective lens in backscattering geometry. Prior to recording measurements, a silicon standard sample exhibiting a single peak at 520 cm^−1^ was used for calibrating the Raman spectrometer.

## 3. Results and Discussion

Figure 1 illustrates the variation of unit-cell volume and calculated density as a function of composition in MAPbBr_3−*x*_Cl*_x_*. It shows a monotonic decrease with increasing Cl content (*x*), highlighting a strong correlation between structural changes induced by Cl substitution and their direct effect on material properties. The reduction in unit-cell volume can be attributed to stronger electrostatic forces resulting from tighter atomic packing when larger Br ions are replaced by smaller Cl ions within the perovskite structure. This can be explained by the higher electronegativity of chlorine (3.16) compared to bromine (2.96), resulting in a stronger covalent character within Pb-Cl bonds compared to Pb-Br bonds. The absolute values of these parameters are tabulated in Table 1. All mixed crystals examined in this study have a cubic phase ($Pm\bar{3}m$) at room temperature [21].

Figure 2 shows the room temperature Brillouin spectra of MAPbBr_3−*x*_Cl*_x_* single crystals and the corresponding fitting lines. The Brillouin spectra consist of two doublets corresponding to the longitudinal (LA) and transverse (TA) acoustic modes. No central peak was observed at room temperature for all compositions. The mode frequencies shift to higher values with increasing Cl content, indicating changes in the elastic properties of the perovskite lattice due to the incorporation of Cl ions. The higher Cl content in the mixed crystal may lead to stronger interatomic interactions, resulting in a stiffer crystal structure, higher elastic moduli and faster acoustic wave propagation. Similar behavior has been observed in previous studies of mixed-halide perovskites, which have shown that the incorporation of halide ions with different ionic radii leads to significant changes in the elastic properties of the materials [12,25].

The acoustic-mode frequencies and the full width at half maximum (FWHM) of the Brillouin doublets were determined by fitting the experimental spectra with the Voigt function, which is a convolution of the instrumental Gaussian and Lorentzian response functions. Considering the backscattering geometry, where the incident and scattered light are antiparallel, the relationship between the acoustic-mode frequency and the sound velocity can be expressed by the following equation:(1)$$V=\frac{\lambda \nu_{B}}{2n}$$
where V denotes the sound velocity, n is the refractive index, λ (=532 nm) is the wavelength of the light source and $\nu_{B}$ is the acoustic-mode frequency determined from experimental spectra by the fitting procedure. The refractive indices of the mixed compositions were estimated by interpolating the experimental values of the pure components, MAPbCl_3_ (n = 1.90) and MAPbBr_3_ (n = 2.30), as shown in Table 1 [24]. The above equation can be used to derive the sound velocities of the LA and TA modes propagating along the cubic [100] direction. According to the Brillouin selection rule, the LA and the TA modes for this scattering geometry are associated with the elastic constants *C*_11_ and *C*_44_, respectively. These elastic constants can then be calculated from the sound velocities and the density (ρ) of the crystal as follows:(2)$$C_{ij}=\rho V^{2}$$

The densities for mixed compositions were also calculated from the lattice parameters obtained and the chemical formula. The densities and refractive indices of all compositions are tabulated in Table 1. Figure 3 shows the calculated LA and TA sound velocities as a function of composition and Figure 4 shows the derived elastic constants *C*_11_ and *C*_44_. It is observed that both LA and TA sound velocities, as well as *C*_11_ and *C*_44_, show a monotonic increase with increasing Cl content. A similar trend was previously observed by Kabakova et al. [12]. Among the two acoustic waves, the changes in V of the LA waves and the related *C*_11_ are much larger than those of the TA waves and *C*_44_.

A key factor contributing to the increase in sound velocity is the reduction in the lattice constant due to the incorporation of the smaller Cl ions, as evidenced by the XRD patterns and lattice parameters discussed above. A stiffer lattice, characterized by higher moduli of elasticity, allows faster propagation of sound waves through the material. Consequently, the observed increase in sound velocity with Cl substitution is expected and supports the hypothesis that Cl incorporation plays a crucial role in modifying the mechanical and acoustic properties of the MAPbBr_3−*x*_Cl*_x_* system.

*C*_11_ is a measure of the stiffness or modulus associated with the resistance of a material to deformation along a primary crystallographic axis when subjected to stress parallel to the same axis, while *C*_44_ represents the shear modulus. Table 2 shows a comparison of the sound velocities and the two elastic constants of the mixed-halide perovskites studied in this work. To the best of our knowledge, the elastic constants for mixed perovskites have not been reported in the literature. For MAPbCl_3_, a theoretical study previously calculated *C*_11_(*C*_44_) to be 39.5 GPa (2.9 GPa), which is close to our experimentally obtained values [26]. These two moduli play an important role in fully characterizing the mechanical behavior of a cubic crystal structure. As *x* increases from 1.5 to 3, *C*_11_ increases from 35.7 GPa to 40.7 GPa, while *C*_44_ increases from 3.63 GPa to 3.71 GPa. While the *C*_44_ value is generally low in all the perovskites studied, it becomes even lower with decreasing Cl content, indicating an increased elastic anisotropy and a low resistance to shear stress in mixed perovskites with lower Cl content [27]. Previously, such a difference has also been observed in MA- and FA-based perovskites [11], as well as in Cs-based perovskites [28], and it has been linked to the rotational motions of the cations and the tilting arrangement of the PbX_6_ octahedra in these halide perovskites [29]. In addition, the observed increase in *C*_11_ and *C*_44_ with higher Cl content suggests that, in the MAPbBr_3−*x*_Cl*_x_* system, a larger amount of chlorine enhances the structural resilience at the molecular level, including both longitudinal and shear deformations, which can help the device withstand the regular strains caused by the external forces during practical operation. However, it should be noted that the change in *C*_11_ is much larger (~12.5%) than that in *C*_44_ (~2%) for the whole composition change, indicating that *C*_44_ is insensitive to the composition of the MAPbBr_3−*x*_Cl*_x_* system.

The absorption coefficient (*α*) was also calculated from the FWHM (Γ*_B_*) of the LA mode by the following relation:(3)$$\alpha =\frac{\pi \Gamma_{B}}{V}$$

Absorption coefficients represent the ability of a material to absorb and lose energy through mechanisms such as thermal vibration and scattering or coupling to other dynamic degrees of freedom [30]. It is therefore a good measure to characterize how acoustic waves are damped in a system during propagation, especially in the gigahertz range in the present case. The calculated absorption coefficients of the LA modes shown in Figure 5 exhibit a decreasing trend with increasing Cl content. The lower absorption/attenuation coefficients observed in materials with higher Cl content suggest that these materials exhibit weaker damping effects or internal friction. The absorption coefficient of the mixed perovskite MAPbBr_1.5_Cl_1.5_ was found to be the highest among all the compositions studied, which may be due to increased lattice heterogeneity leading to attenuation of acoustic wave propagation. Especially, the heterogeneous distribution of Br and Cl cations in the X site may be enhanced near MAPbBr_1.5_Cl_1.5_ where the ratio between Cl and Br is 1:1 maximizing the damping of the acoustic waves. The above observations highlight the importance of investigating how compositional variations not only affect mechanical properties, but also influence energy dissipation mechanisms within the MAPbBr_3−*x*_Cl*_x_* mixed-halide perovskite system. On the other hand, the absorption coefficients of the TA modes were much higher than those of the LA modes, more than 9 × 10^5^ m^−1^. In addition, the absorption coefficient of the TA mode was insensitive to the composition, consistent with the fact that the corresponding elastic constant *C*_44_ is insensitive to the composition. This suggests that the TA modes are more strongly damped in the mixed crystals compared to the LA modes, which are common to all mixed crystals regardless of composition.

Investigation of the bonding mechanism between the halides (Cl and Br) and the surrounding octahedra (PbCl_6_ and PbBr_6_) may be necessary, as the elastic properties of halide perovskites have been found to be determined by the strength of the chemical bond between the Pb and halogen atoms [31,32]. Raman spectroscopy serves as a useful tool to obtain important information about the vibrational properties and molecular dynamics within these mixed-halide perovskites [22]. Figure 6 shows the Raman spectra recorded at room temperature for all compositions in the frequency range 10–3500 cm^−1^. The Raman spectra show several distinct peaks corresponding to vibrational modes of the lattice as well as the internal modes of the MA cation. The exact peak positions for all the observed vibrational modes, calculated by fitting analyses, together with their respective mode assignments, are tabulated in Table 3 based on previous results [23,33,34].

Figure 6a shows the low-frequency lattice modes below 300 cm^−1^ and the torsional (τ) mode of the MA cation. We observed a significant increase in mode frequencies for the low-frequency lattice modes and the torsional (τ) mode of the MA cation with increasing Cl content. A similar trend was also observed by Xie et al. for a small increase in Cl content where the authors studied MAPbBr_3−*x*_Cl_3_ with *x* = 0.25, 0.33 and 0.5 [22]. Low-frequency lattice modes represent collective vibrations of the crystal lattice, including motions associated with PbX_6_ octahedral arrangements (where X refers to Br or Cl). Two low-frequency modes associated with the bending and stretching of the lead-halide octahedra increase from 55 cm^−1^ and 151 cm^−1^ in MAPbBr_1.5_Cl_1.5_ to 77 cm^−1^ and 178 cm^−1^ in MAPbCl_3_, respectively. As discussed earlier, the incorporation of smaller Cl ions into the lattice results in a decrease in unit-cell dimensions, leading to stronger electrostatic interactions between neighboring atoms. Consequently, these stronger interactions imply stiffer force constants within the crystal structure, which translates into increased vibrational frequencies of low-frequency lattice modes.

The torsional (τ) mode, which corresponds to an out-of-phase twisting motion around the central nitrogen atom in MA cations, increases from 453 cm^−1^ to 484 cm^−1^ as *x* increases from 1.5 to 3. This mode is sensitive to hydrogen bonding and the overall interaction strength between the organic moiety and the surrounding halogen atoms [35]. With increasing Cl content, stronger hydrogen bonding may occur due to shorter bonding distances between the amine end (NH_3_^+^) of the MA cation and the corresponding halogen atoms, ultimately resulting in a shift of the τ mode peak towards higher wavenumbers. These results can be correlated with our Brillouin spectroscopic observations of the elastic properties. The linear translational–rotational coupling of the MA cation has been found to cause a softening of the perovskite lattice, as well as a damping effect on sound waves propagating within the crystal [36]. As the MA cations undergo rotational and translational motions, they impose local strain on the adjacent inorganic lattice. This local strain affects the interatomic forces between neighboring atoms, resulting in variations in the elastic constants and mechanical stiffness of the perovskite lattice.

Figure 6b shows Raman spectra between 800 and 1800 cm^−1^, including rocking modes of the MA cation and bending of CH_3_ and NH_3_ groups. Contrary to the behavior of the low-frequency modes, the vibrational modes in this range show no significant change in Raman shifts as a function of composition. Similar behavior was observed for high-frequency modes involving internal bond vibrations such as C-H and N-H stretching within the organic cations, as shown in Figure 6c. The insensitivity of these modes to Cl content implies a minimal influence of halide substitution on the local environment surrounding the organic component and suggests that Cl content primarily affects global crystal-lattice dynamics, while only moderately or negligibly affecting local bond strengths within the organic cations.

The softening of the lattice structure makes the crystal more susceptible to local distortions that can lead to the formation of polarons, which can ultimately have a significant impact on the electronic and optical properties of a material in photovoltaic devices, as they tend to lower the carrier mobility due to their increased effective mass and limited hopping behavior compared to free carriers [27,37,38]. On the other hand, a soft lattice structure can be easily damaged by external shock or vibration, reducing long-term mechanical reliability. Therefore, understanding the elastic behavior of halide perovskites is crucial for the development of advanced and reliable perovskite materials with tailored mechanical and functional properties for use in various optoelectronic and photovoltaic applications.

## 4. Conclusions

The study of mixed-halide perovskite MAPbBr_3−*x*_Cl*_x_* single crystals revealed the relationship between halogen content and the mechanical, acoustic and vibrational properties in this system. X-ray powder diffraction showed a decrease in the lattice parameter with increasing Cl content, which was attributed to the smaller ionic radii of Cl compared to Br. Brillouin spectroscopy showed that higher Cl incorporation resulted in higher sound velocities and elastic moduli, resulting in a stiffer crystal structure that allowed faster propagation of acoustic waves through the material. Raman spectroscopy provided further information on how Cl content affects the vibrational properties and molecular dynamics within the mixed-halide perovskites. Low-frequency lattice modes and torsional modes of the MA cation showed significant increases in wavenumber with increasing Cl content, highlighting stronger electrostatic interactions and hydrogen bonding within the lattice. In contrast, vibrational modes involving rocking, bending and high-frequency internal bond vibrations within the organic component were insensitive to the Cl substitution, suggesting that Cl content primarily affects global crystal-lattice dynamics while only moderately or negligibly affecting local bond vibrations within organic cations.

Elasticity of the perovskite layer is crucial for thin-film technology in perovskite solar cells on flexible substrates, as it prevents cracking and performance degradation. Single crystalline layers are preferrable since the mechanical behavior of polycrystalline layers can be compromised by grain boundaries and defects. Understanding the relationships between compositional variations and the mechanical, acoustic and vibrational properties is crucial for the development of advanced perovskite materials with tailored properties for various optoelectronic and photovoltaic applications. The present results can serve as fundamental physical properties for the incorporation of mixed-halide systems in various optoelectronic applications.

## Figures and Tables

**Figure 1 materials-16-03986-f001:**
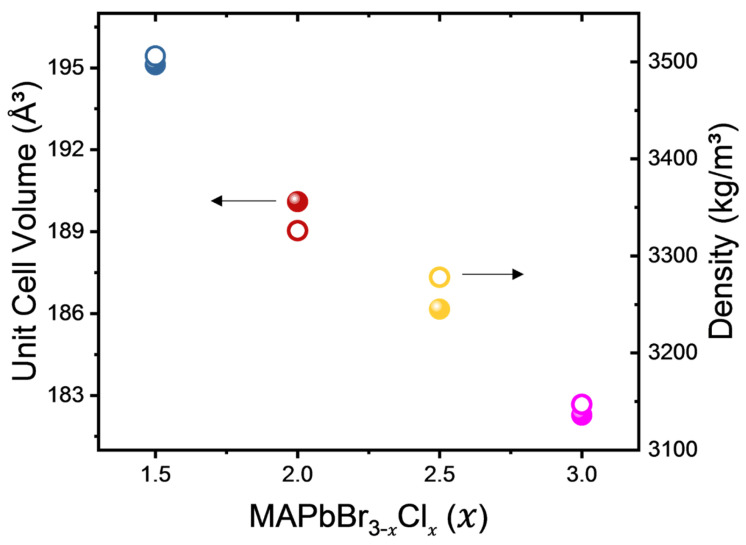
Unit-cell volume (spheres) and density (circles) of MAPbBr_3−*x*_Cl*_x_* with *x* = 1.5, 2, 2.5 and 3.

**Figure 2 materials-16-03986-f002:**
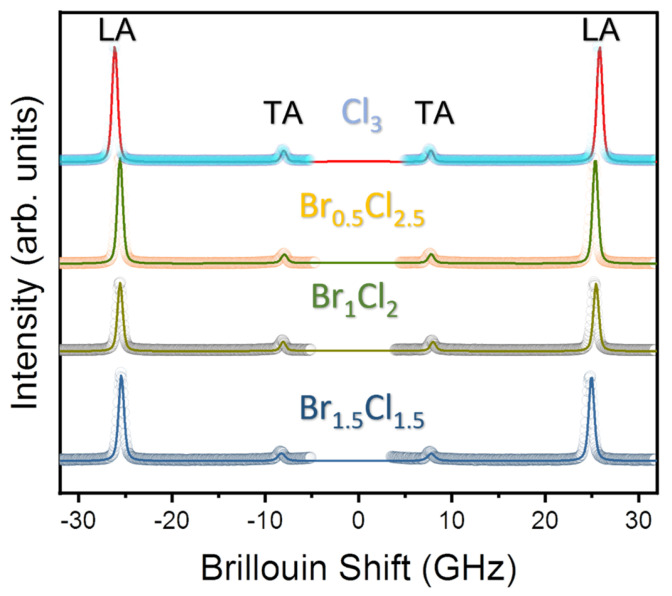
Room-temperature Brillouin spectra of the four mixed-halide perovskites MAPbBr_3−*x*_Cl*_x_* with *x* = 1.5, 2, 2.5 and 3 along with best fitted lines using Voigt function.

**Figure 3 materials-16-03986-f003:**
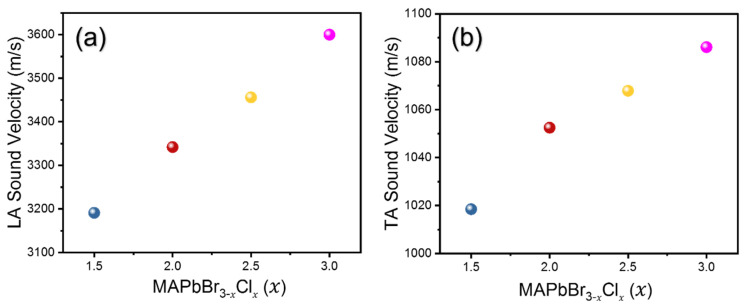
(**a**) Longitudinal acoustic (LA) and (**b**) transverse acoustic sound velocities as a function of composition *x* in MAPbBr_3−*x*_Cl*_x_*.

**Figure 4 materials-16-03986-f004:**
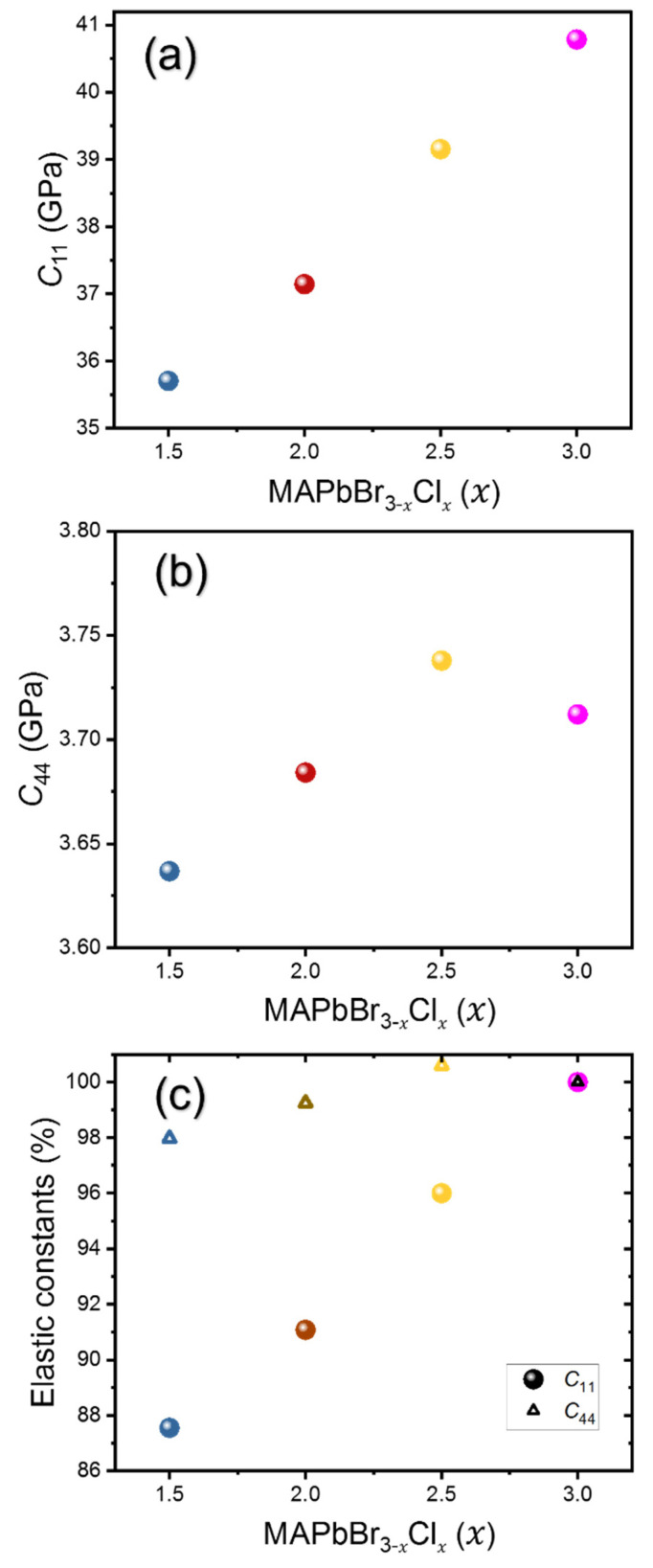
Two elastic constants, (**a**) *C*_11_ and (**b**) *C*_44_, and (**c**) their relative changes with respect to the values of MAPbCl_3_ as a function of composition *x* in MAPbBr_3−*x*_Cl*_x_*.

**Figure 5 materials-16-03986-f005:**
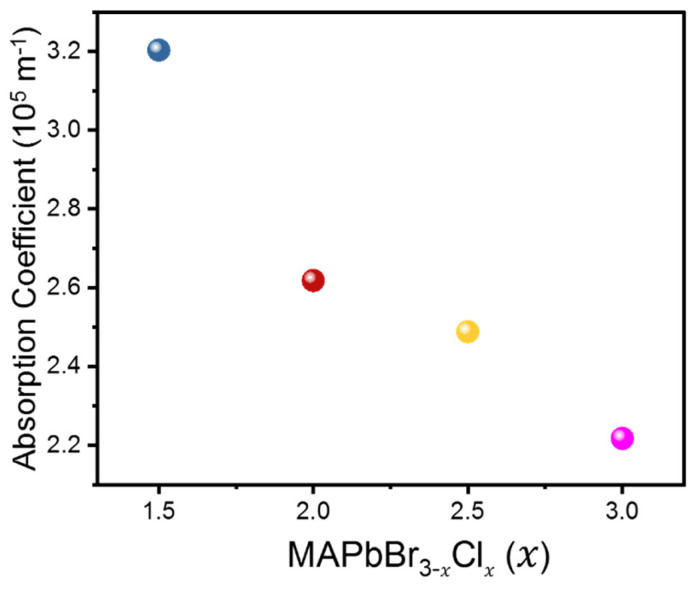
Absorption coefficient of the LA mode as a function of composition *x* in MAPbBr_3−*x*_Cl*_x_*.

**Figure 6 materials-16-03986-f006:**
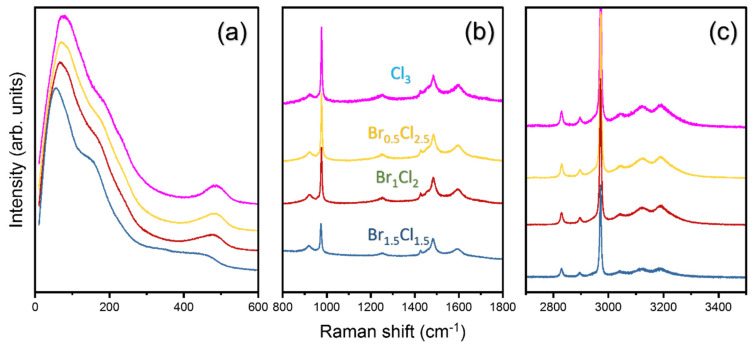
Room-temperature Raman spectra of MAPbBr_3−*x*_Cl*_x_* with *x* = 1.5, 2, 2.5 and 3 in (**a**) 10–600 cm^−1^, (**b**) 800–1800 cm^−1^ and (**c**) 2700–3500 cm^−1^ ranges.

**Table 1 materials-16-03986-t001:** Summary of crystallographic and physical properties for MAPbBr_3−*x*_Cl*_x_* with *x* = 1.5, 2, 2.5 and 3. Values of refractive index for pure components were obtained from ref [24] and interpolated to get values for all mixed compositions.

Composition	Lattice Parameter (Å)	Unit-Cell Volume (Å³)	Refractive Index	Density (kg/m^3^)
MAPbCl_3_	5.67	182.28	1.90	3147
MAPbBr_0.5_Cl_2.5_	5.71	186.16	1.96	3278
MAPbBr_1_Cl_2_	5.75	190.10	2.03	3326
MAPbBr_1.5_Cl_1.5_	5.80	195.11	2.10	3506

**Table 2 materials-16-03986-t002:** Summary of calculated sound velocities and elastic constants for MAPbBr_3−*x*_Cl*_x_* with *x* = 1.5, 2, 2.5 and 3.

Composition	*V*_LA_ (m/s)	*V*_TA_ (m/s)	*C*_11_ (GPa)	*C*_44_ (GPa)
MAPbCl_3_	3599	1086	40.7	3.71
MAPbBr_0.5_Cl_2.5_	3455	1067	39.1	3.73
MAPbBr_1_Cl_2_	3341	1052	37.1	3.68
MAPbBr_1.5_Cl_1.5_	3191	1018	35.7	3.63

**Table 3 materials-16-03986-t003:** Raman modes along with their respective assignments for MAPbBr_3−*x*_Cl*_x_* with *x* = 1.5, 2, 2.5 and 3.

Br_1.5_Cl_1.5_	BrCl_2_	Br_0.5_Cl_2.5_	Cl_3_	Assignment
55	65	73	77	δs (X-Pb-X) ^a^
151	161	168	178	ν_s_ (Pb-X) ^a^
226	227	227	236	R of MA+ cation ^a^
453	468	479	484	τ (MA) ^a,b^
920	923	923	922	ρ (MA) ^a,b^
974	975	976	976	ν (C-N) ^a,c^
1248	1248	1248	1247	ρ (MA) ^a,c^
1426	1426	1426	1426	δ_s_ (CH_3_) ^a^
1454	1456	1456	1456	δ_as_ (CH_3_) ^a^
1483	1484	1485	1484	δ_s_ (NH_3_) ^a^
1593	1596	1596	1596	δ_as_ (NH_3_) ^a^
2829	2830	2830	2830	ν_as_ (C-H) ^c^
2895	2896	2897	2897	ν_s_ (C-H) ^c^
2870	2971	2972	2972	ν_as_ (CH_3_) ^a^
3041	3042	3043	3044	ν_s_ (NH_3_) ^a^
3115	3116	3117	3117	ν_as_ (CH_3_) ^b^
3190	3192	3194	3196	ν_s_ (NH_3_) ^b^

δ: bending; ρ: rocking; ν: stretching; s/as: symmetric/asymmetric; τ: torsion; R: rotation. Superscripts ^a^, ^b^ and ^c^ indicate the references [23,33,34], respectively, from which the mode assignment was adapted.

## Data Availability

The data in this study is available upon request to the corresponding author.

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
