# Peer review of "Influence of Halides on Elastic and Vibrational Properties of Mixed-Halide Perovskite Systems Studied by Brillouin and Raman Scattering"

_materials, 2023, doi:10.3390/ma16113986_

Round 1

Reviewer 1 Report

The relationship between halogen content and mechanical, acoustic and vibrational properties of mixed halide perovskite MAPbBr3-xClsingle crystals was investigated.  This manuscirpt is well written, I have one question: Which x of MAPbBr3-xCldo you recommend for different  optoelectronic and photovoltaic applications ?

Author Response

We appreciate the reviewers’ critical reading and valuable opinions on our manuscript. In the original manuscript, changes were made using “Track changes: option in MS word.

  1. The relationship between halogen content and mechanical, acoustic and vibrational properties of mixed halide perovskite MAPbBr3-xClsingle crystals was investigated.  This manuscript is well written, I have one question: Which x of MAPbBr3-xClx  do you recommend for different  optoelectronic and photovoltaic applications ?
  • Thank you very much for the reviewer’s favorable opinion on our manuscript. The x value in MAPbBr3-xClx significantly affects material properties for optoelectronic or photovoltaic applications. In photovoltaics, lower x values favor higher efficiency due to smaller bandgap but may compromise stability. For optoelectronics, x selection depends on desired wavelength range or color emissions. Balancing performance and stability is essential when optimizing halide composition for specific application requirements. In the manuscript, as it is mentioned in the introduction second last line : “In addition, mixed perovskite materials offer versatility through their compositional flexibility, allowing researchers to tailor their properties according to the desired application.”

Reviewer 2 Report

I have reviewed the manuscript entitled "Influence of Halogens on Elastic and Vibrational Properties of Mixed Halide Perovskite Systems Studied by Brillouin and Raman Scattering" submitted to Materials for publication in the special issue 100th Anniversary of Brillouin Scattering.  While it is my task to evaluate and criticize the work to uphold a level of quality and rigor, I feel that the manuscript is in fact rather well written and polished in its original form.  The results are quite interesting and relevant to both the special issue concerned as well as the Materials community in general.  Rather trivially, I discovered a single typo to indicate for improvement.  Before the reference [21] in the text, the formula MAPbr3-xCl3 is missing a "B".  Despite this, I can confidently suggest the manuscript for publication as it is.

Author Response

We appreciate the reviewers’ critical reading and valuable opinions on our manuscript. In the original manuscript, changes were made using “Track changes: option in MS word.

I have reviewed the manuscript entitled "Influence of Halogens on Elastic and Vibrational Properties of Mixed Halide Perovskite Systems Studied by Brillouin and Raman Scattering" submitted to Materials for publication in the special issue 100th Anniversary of Brillouin Scattering.  While it is my task to evaluate and criticize the work to uphold a level of quality and rigor, I feel that the manuscript is in fact rather well written and polished in its original form.  The results are quite interesting and relevant to both the special issue concerned as well as the Materials community in general.  Rather trivially, I discovered a single typo to indicate for improvement.  Before the reference [21] in the text, the formula MAPbr3-xCl3 is missing a "B".  Despite this, I can confidently suggest the manuscript for publication as it is.

  • Thank you very much for the reviewer’s favorable opinion on our manuscript. The mentioned typo was corrected in the manuscript.

Reviewer 3 Report

Structural, vibrational (Raman) and elastic (Brillouin) properties are measured in the MAPbBr1-xClx series. The sound velocity (elastic) measurements are reported for the first time, while the Raman measurements extend previous measurements of the same authors for lower frequencies. The shift of Raman peaks at lower values is related to the translation-rotational vibration of MA molecules, and its increase with Cl is related to the variation in bonding and elastic properties.

I have detected no errors, and the presentation of the results is clear. I support the publication of this manuscript in its current form.

Author Response

Thank you very much for the favorable opinion on our manuscript. 

Reviewer 4 Report

The submission presents an experimental study of the mixed halide perovskite MAPbBr3-xClx single crystals for identification of the relationship between halogen content and mechanical, acoustic and vibrational properties in these crystals. The topis is new and to some extent interesting in view of application of perovskites in photovoltaics and optoelectronics.

Elasticity of perovskite layer is important for thin film technology of perovskite solar cells deposited on various not stiff substrates. This should be mentioned in the paper and some comment on the relationship of mechanical properties of single crystals and of polycrystalline layers utilized in solar cells should be added.

Thus, the phrase in the Conclusions, 

"Understanding these relationships between compositional variations and mechanical, acoustic, and vibrational properties are crucial for the development of advanced perovskite materials with tailored properties for various optoelectronic and photovoltaic applications." should be developed into a separated paragraph.

Similarly the short presentation of perovskites in application to photovoltaics in the beginnig of the Introduction is too optimistic and thus slightly misleading. Limits for the utilization of perovskite cells are stronger and presented more realistically e.g., in Materials 2022, 15, 2254. https://doi.org/10.3390/ma15062254. The references should be thus developed with mentioning about main obstacles on the way for wider usability of perovskite photovoltaics. 

Some more precise presentation is required, in particular, the phrase,

"We study the variations in optical mode and acoustic-mode frequencies as a function of halide composition, where the chlorine content increases from 1.5 to 3 at the X site"

is unclear -- what does mean X here;

the proof-reading ot the whole text  is also required;

the revised version could be considered for publication in Materials.

English formulation is not perfect and a thorough editing and proof-reading is required 

Author Response

We appreciate the reviewers’ critical reading and valuable opinions on our manuscript. In the original manuscript, changes were made using “Track changes: option in MS word.

  1. Elasticity of perovskite layer is important for thin film technology of perovskite solar cells deposited on various not stiff substrates. This should be mentioned in the paper and some comment on the relationship of mechanical properties of single crystals and of polycrystalline layers utilized in solar cells should be added. Thus, the phrase in the Conclusions, "Understanding these relationships between compositional variations and mechanical, acoustic, and vibrational properties are crucial for the development of advanced perovskite materials with tailored properties for various optoelectronic and photovoltaic applications." should be developed into a separated paragraph.
  • Thank you very much for the reviewer’s favorable opinion on our manuscript. The said improvements were done in the conclusion part as: “Elasticity of the perovskite layer is crucial for thin-film technology in perovskite solar cells on flexible substrates, as it prevents cracking and performance degradation. Single crystalline layers are preferrable since the mechanical behavior of polycrystalline layers can be compromised by grain boundaries and defects.” Conclusion (2nd paragraph)
  1. Similarly the short presentation of perovskites in application to photovoltaics in the beginnig of the Introduction is too optimistic and thus slightly misleading. Limits for the utilization of perovskite cells are stronger and presented more realistically e.g., in Materials 2022, 15, 2254. https://doi.org/10.3390/ma15062254. The references should be thus developed with mentioning about main obstacles on the way for wider usability of perovskite photovoltaics. 
  • Amendments in the beginning of the introduction part were made as asked. The reference was studied and was cited to add further to introduction part in the last line of first paragraph as: “Despite the numerous advantageous properties of perovskites, their long-term stability remains a significant limitation that requires further investigation [9].” Introduction (1st paragraph)

  1. Some more precise presentation is required, in particular, the phrase,"We study the variations in optical mode and acoustic-mode frequencies as a function of halide composition, where the chlorine content increases from 1.5 to 3 at the X site” is unclear -- what does mean X here.
  • The sentence was extended (red part) to provide a clear meaning as follows: “We study the variations in optical-mode and acoustic-mode frequencies as a function of halide composition, where the chlorine content increases from 1.5 to 3 at the X site anion in the ABX3 perovskite structure.” Introduction (3rd paragraph)
  1. The proof-reading of the whole text  is also required.
  • The manuscript was proof-read thoroughly.